# Alpha Radiation-Induced Luminescence by Am-241 in Aqueous Nitric Acid Solution

**DOI:** 10.3390/s19071602

**Published:** 2019-04-02

**Authors:** Thomas Kerst, Rikard Malmbeck, Nidhu lal Banik, Juha Toivonen

**Affiliations:** 1Photonics Laboratory, Physics Unit, Tampere University, P.O. Box 692, 33101 Tampere, Finland; juha.toivonen@tuni.fi; 2Helsinki Institute of Physics, Helsinki University, P.O. Box 64, 00014 Helsinki, Finland; 3European Commission, Joint Research Centre (JRC), Directorate G–Nuclear Safety and Security, Advanced Nuclear Knowledge, 76125 Karlsruhe, Germany; rikard.malmbeck@ec.europa.eu (R.M.); nidhu.banik@ec.europa.eu (N.B.)

**Keywords:** alpha radiation, radioluminescence, liquid phase luminescence, americium

## Abstract

When exposed to air, alpha particles cause the production of light by exciting the molecules surrounding them. This light, the radioluminescence, is indicative of the presence of alpha radiation, thus allowing for the optical sensing of alpha radiation from distances larger than the few centimeters an alpha particle can travel in air. While the mechanics of radioluminescence in air and other gas compositions is relatively well understood, the same cannot be said about the radioluminescence properties of liquids. Better understanding of the radioluminescence properties of liquids is essential to design methods for the detection of radioactively contaminated liquids by optical means. In this article, we provide radioluminescence images of Am-241 dissolved in aqueous nitric acid (${\mathrm{HNO}}_{3}$) solution and present the recorded radioluminescence spectrum with a maximum between 350nm and 400nm, and a steep decrease at the short wavelength side of the maximum. The shape of the spectrum resembles a luminescence process rather than Cerenkov light, bremsstrahlung, or other mechanisms with broadband emission. We show that the amount of light produced is about 150 times smaller compared to that of the same amount of Am-241 in air. The light production in the liquid is evenly distributed throughout the sample volume with a slight increase on the surface of the liquid. The radioluminescence intensity is shown to scale linearly with the Am-241 concentration and not be affected by the ${\mathrm{HNO}}_{3}$ concentration.

## 1. Introduction

Radioluminescence describes the spontaneous emission of light as a consequence of interaction of luminescent material with ionizing radiation. The ionizing particle typically originates from, but is not limited to, a form of radioactive decay. In the process of creating radioluminescence, the ionizing particle or an induced secondary electron collides with a luminescent material, resulting in the elevation of an orbital electron [1]. The excited electron then has a chance to radiatively decay, thereby emitting a photon of light. Photons created this way are said to be radioluminescence, since the presence of ionizing radiation ultimately lead to its production. Thus, the presence of radioluminescence is to be taken as an indicator for the presence of ionizing radiation itself.

Radioluminescence induced by alpha particles allows for their remote detection by optical means. In air, alpha particles come to a halt after having travelled for about 4cm, losing almost all their kinetic energy in the process [2]. In contrast, beta and gamma radiation travel a few meters and tens of meters in air, respectively [3], making alpha radiation a comparably short ranged type of nuclear radiation. This makes it rather easy to avoid exposure to a known alpha source by keeping distance, while at the same time making it difficult to detect unknown alpha sources with methods that rely on direct interaction with the particles. Alpha induced radioluminescence is not limited to a travel distance of a few cm, thus making it possible to remotely detect alpha radiation by collecting radioluminescent light. [4].

How little alpha radiation can reliably be detected by collecting radioluminescence depends on the conditions surrounding the source. To understand the reasons for this, it is necessary to recognize that even high-activity alpha sources create amounts of photons that can be considered almost negligible in comparison to the number of photons a typical ambient light source, such as the sunlight on earth or a fluorescent door sign, emits. A single alpha particle being stopped in air leads to the production of about 100 photons [5,6], each of which is isotropically radiated away from the location of the alpha source. Thus, photon counting devices must be employed to make radioluminescence detectable. In a recent study, Sand et al. [7] thoroughly investigated the question of how small an alpha contamination can one detect by optical means alone, given that one can use state-of-the art equipment to detect single UV photons and the site of contamination is exposed to normal air. Under ideal conditions, e.g., the only present light source being radioluminescence, they were able to detect sources with an alpha activity of 4kBq or surface contamination with an activity of $300\;\mathrm{Bq}\;{\mathrm{cm}}^{-2}$, respectively, by using an UV sensitive low-noise PMT at a distance of about 1m with a measurement time of about 10s.

Once background lighting must be taken into account, a straightforward approach of collecting light is no longer feasible. The spectral patterns of both the air radioluminescence, being mostly made up of $N_{2}$ emissions [1], and of the background lighting must be considered. Furthermore, the optical design must be optimized to mitigate the effects of the typically high ambient light levels [7,8,9,10,11]. Another way of overcoming the problem of background lighting is by only collecting photons with wavelengths that are in the UVC wavelength range at wavelengths shorter than 280nm. Daylight, one of the most common source background lighting containing UV parts, does not extend into the UVC [12]. Thus, choosing equipment that is both sensitive to those wavelengths and able to detect single photons can be used for remote detection of alpha radiation. Ivanov et al. [13,14] are one of the first reporting on such an approach using cameras, whereas Crompton et al. [15] pioneered the field with using a detector that uses the photoelectric effect and gas multiplication to detect individual UVC photons. However, restricting the detection to the UVC limits the sensitivity of the optical detection as UVC range contains only a small fraction of the radioluminescence [16]. One way to overcome this impediment is by relying, when possible, on an artificial atmosphere that allows for enhanced yield of radioluminescence [17,18,19].

The ability of liquids to exhibit luminescence upon irradiation with ionizing particles has not been as thoroughly studied as it has been done for gases. Even though there have been a few studies reaching back into the 60s [20], renewed interest into the topic has been partly sparked by the observations of Yamamoto and colleagues [21]. They showed that faint radioluminescence is produced by placing an Am-241 source in water. In a later study, it was estimated using long-pass filters that the spectrum is a very broad one with increasing intensity towards shorter wavelengths [22]. However, a fully resolved spectrum was not presented.

In this paper, we present radioluminescence images originating from a liquid with Am-241 dissolved in an aqueous solution of nitric acid (${\mathrm{HNO}}_{3}$). The imaging reveals equal distribution of light production throughout the sample volume with a slight increase at the surface of the liquid interfacing with air. We present a fully resolved radioluminescence spectrum in the spectral range between 280nm and 550nm. Lastly, we investigate the intensity of the radioluminescence as a function of the amount of Am-241 and the ${\mathrm{HNO}}_{3}$ concentration, and discuss about the possible mechanisms producing the observed radioluminescence.

In Section 2 we describe our experimental arrangements and give detailed information on how data was obtained. We also given detailed information about the image processing involved and how the reader can reproduce the algorithms and methods used in the digital post-processing. The results are presented and discussed in Section 3. Section 4 concludes this work and summarizes the main findings.

## 2. Materials and Methods

All experiments have been carried out in the laboratories and premises of the Joint Research Centre (JRC) in Karlsruhe, Germany, during two measurement campaigns. For safety reasons, all radioactive material used was kept and handled in a plexiglass glovebox while most of the other equipment remained outside. The only modifications to the glovebox were a replacement of one glove with a quartz window (Sico Technology GmbH, SQ1, Austria) and the insertion of another quartz glass window into the plexiglass ceiling of the box. The chosen quartz glass has a flat transmission spectrum down to 180nm wavelength and is thus well suited for this experiment [23].

The box has been shielded from ambient light with black rubberized fabric (Thorlabs, BK5, Newton, NJ, USA). A very thorough shielding with multiple layers was required to block all ambient light, especially given that the laboratory could not be made totally dark. We made sure that radioluminescence was the only significant source of light during the data acquisition by recording a control image without the alpha source prior to actual data recording with different radioactive samples.

The radioactive samples used during the measurements included a planchet with a thin layer of Pu-239 evaporated onto it (from now on referred to as: the solid sample) and several liquids containing Am-241 (from now on referred to as: the liquid samples). The surface activity of the solid sample was verified by alpha spectroscopy to be 5.6MBq. We prepared the liquid samples by evaporating a pure Am-241 solution to dryness and then adding nitric acid (${\mathrm{HNO}}_{3}$) solution to it. Depending on the sample, we used 1M, 3M or 7M, with the solvent having been deionized water. M is the molar concentration, i.e., mols per liter. We prepared each liquid sample to measure a volume of 2mL. We denote the concentration of Am-241 dissolved in each sample by dividing the amount of μg of Am-241 with the number of grams of the liquid sample. Thus, the concentration is defined by mass, and throughout the paper we express it with the unit ppm. We prepared a total of 7 liquid samples divided on 2 distinct sets. The first set comprised a total of 3 different solutions each containing 150ppm of Am-241, but with varying ${\mathrm{HNO}}_{3}$ concentrations of 1M, 3M and 7M. The second set comprised a total of 4 different solutions each with a ${\mathrm{HNO}}_{3}$ concentration of 3M, but with varying Am-241 concentrations being 50ppm, 75ppm, 150ppm and 750ppm. Using the specific activity of Am-241 [24], the liquid samples have activities of roughly 13MBq, 19MBq, 38MBq and 190MBq, respectively. When using a liquid sample in an experiment, we filled it into a standard quartz glass fluorescence cuvette that transmits light at the wavelength range of 200nm–2500nm (12.5mm length and width, 45mm height, 10mm light path length, Hellma analytics GmbH & Co. KG, type: 101-QS, material: QS).

### 2.1. Radioluminescence Imaging

We imaged the radioluminescence by placing a sample on a holder close to the quartz window and an EMCCD camera (Andor iXon3 897) on the other side of the window. Sample holder and camera objective were kept at a constant distance of about 30cm throughout all experiments. The camera sensor was internally cooled down to −80 °C to reduce thermal noise. We placed the sample and the holder insight a light-tight PVC tube and wrapped the camera in black fabric to further improve the ambient light shielding. The imaging lens was a UV-objective (Universe Kogaku, UV1228CM) with a focal length of 12mm and a light collection efficiency of f/2.8. A schematic illustration of the arrangement is shown in Figure 1.

For each sample we took one background image in normal laboratory lighting conditions and then darkened the setup to obtain images of the radioluminescence. For each liquid sample we took 30 images with an exposure time of 60s each, resulting in a total integration of 30min. For the solid sample, a single exposure of 100s length proofed sufficient to acquire image material with good enough signal-to-noise ratio. The individual images accumulated numerous hot pixels, e.g., pixels with maximum value, which likely occurs due to exposure of the sensor to gamma radiation predominantly from the 59.6keV line of Am-241.

All image material has been post-processed by taking the pixel-wise median of the 30 consecutive images. The so-constructed median image is largely free of hot pixels all the while leaving the features and resolution of the image unaffected. The procedure is illustrated in Figure 2 with data from the experiments serving as example. For image processing we used python 3.6.3 as programming language and made use of the libraries OpenCV 3.3.1, matplotlib 2.1.0 and NumPy 1.13.3.

### 2.2. Spectral Measurement

The radioluminescence spectrum was measured by transporting light with a system of liquid light guides outside the glove box, where it was analyzed. A schematic illustration of the arrangement is presented in Figure 3. The sample was placed on a height-adjustable pedestal inside the glovebox. The liquid light guide (Lumatec GmbH, Series 300, 8mm core diameter) was installed next to the pedestal to collect the light emanating from the sample. The fiber was put as close as possible to the sample to eliminate the need for collimating optics. When the liquid sample was spectrally analyzed, the fiber was installed as close as possible to the sample holder to make sure that only radioluminescence from the bulk of the liquid reaches the detector. From there the light was guided to a 10mm thick quartz window in the ceiling of the glove box. Another liquid light guide at the other side of the window then picked up the light and further transported it to a motorized monochromator (Horiba Scientific Inc., iHR 550), where it was spectrally dispersed. Light of only one selected wavelength leaves the monochromator at a time. The monochromator was operated with a UV sensitive grating (model 51050, $300\;\mathrm{gr}\;{\mathrm{mm}}^{-1}$, 250nm blaze) and an entrance slit width of 2mm, resulting in a spectral resolution of about 6nm. The light passing the monochromator was measured with a low-noise photon counting PMT (Perkin Elmer GmbH, MP-1982, <1cps DC) and read out via computer.

A spectrum was obtained by counting the photons with the PMT at each individual wavelength and then arranging the so-measured counts according to their associated wavelength. The monochromator scanned from 280nm to 550nm in 3nm steps, integrating for a total of 10s at each step. The scan was repeated multiple times both to improve the signal-to-noise ratio by having more data points available and to compensate for drifts of the PMT. A drift of the PMT could not have been counteracted against if the measurement consisted of one long single scan.

During post-processing, the spectral data we accounted for effects that alter the measured spectrum. The cuvette, the light guide, and the PMT have a flat response. They can change the intensity of the light they respond to, but they cannot alter the shape of the spectrum of this light. The reflectance of the grating has been calibrated to give a flat spectral response. A 10mm thick water column has a negligible absorption in the measurement wavelength range [25], thus it has no effect on the measured spectral shape.

## 3. Results and Discussion

### 3.1. Imaging

Radioluminescence of 750ppm Am-241 which has been dissolved in 3M
${\mathrm{HNO}}_{3}$ was imaged with the EMCCD camera in darkness. In addition, we took a photograph image under normal lighting conditions for a reference. We repeated the procedure for the Pu-239 solid sample planchet, where we expected radioluminescence of air to occur above it. That served as an intensity calibration regarding to previously published results [26]. The recorded images are shown in Figure 4.

The image of the solid planchet radioluminescence in Figure 4b shows radioluminescence emanating from the surface of the planchet, in a fashion similar to what has been found and described in greater detail by Sand et al. in an experiment under similar conditions [26]. The imaging of radioluminescence emanating from the liquid was performed with a total integration time of 1800s, whereas the imaging the solid sample was performed in 100s. This is due to the much lower light yield of the liquid, even though the overall alpha activity is higher. We estimated the light yield per alpha particle in the liquid by integrating the pixel intensities of the background-corrected planchet image and dividing with activity and exposure time. A similar procedure has been used for to the central parts of liquid radioluminescence image, and the resulting light yield was further scaled to take into account the total liquid area in the cuvette. The energies of alpha particles emitted by Pu-239 are 5.157MeV and by Am-241 are 5.486MeV, respectively [27], thus the two samples are quite similar from an alpha radiation point of view. From this analysis, we find that the light yield of an alpha particle in the liquid is roughly 150 times lower than in air. A similar effect has been noticed by Yamamoto et al. in their research on liquid radioluminescence [21], where they observe a decreased light yield with a factor of about 100.

The pixel intensities at the liquid-air boundary in Figure 4d are higher compared to the pixel intensities further down the water column. The shape of this area is coincidental with the shape of the liquid-air boundary in Figure 4c, which can be even better seen in the overlay image in Figure 4e. The camera is slightly tilted towards the cuvette, which makes it possible to look on top of the interface of the liquid-air boundary. The increase in the intensity might be due to an optical effect or an increased light production at the surface. The optical refraction at the liquid-air boundary can redirect more radioluminescent light towards the camera. Also, it is known that alpha particles are emitted out from the liquid [28], and thus can create radioluminescence in air [5]. The radioluminescence yield in air was previously shown to be approximately 150 times more efficient, which can result in the increased intensity at the surface in Figure 4d.

It can also be seen there is an area of slightly increased pixel intensities at the lower end of the cuvette. The camera looks with a slightly tilted angle at the cuvette, making a part of the cuvette which is surrounded by the holder visible to the camera. It is those parts of the image that show increased pixel intensities. In the overlay image in Figure 4e this becomes even clearer. Thus, it is very likely that the increased intensity in the radioluminescence image in this particular area is not created by an increased radioluminescence production but rather by reflection of radioluminescence photons from the sample holder towards the camera.

Vertical and horizontal cross sections of the radioluminescence image of the cuvette are shown in Figure 5 for detailed analysis of the intensity distribution. In the horizontal cross section in Figure 5a it can be seen that the radioluminescence is evenly distributed throughout the liquid. We notice that the glass walls have no influence on the light production. In the vertical cross section in Figure 5b the differences in radioluminescence intensity as a function of height become clearly visible. The rightmost peak corresponds to the slightly increased intensity seen at the point where the cuvette is submerged in the holder. The maximum in the vertical cross section shows the increased light production at the surface of the liquid. Dotted lines indicate the position where the interface of the liquid-air boundary faces the camera. They correspond to the same dotted lines shown in Figure 4. Between these two peaks, the radioluminescence intensity is notably evenly distributed.

The local maxima to the left in Figure 5b correspond to the increased radioluminescence observed at the top end of the cuvette. This is very likely radioluminescence from dried liquid. The same cuvette was used to hold all liquid samples. Changing the sample made it necessary to empty and refill the cuvette. During refilling contact of the liquid with the walls was difficult to avoid, thus contaminating it. Even careful cleaning rarely removed all contamination, and it were the edges of the cuvette that proved most difficult to clean. It is this contamination that is seen in the radioluminescence image. Though the radioluminescence from contamination of the cuvette walls is somewhat of a nuisance for this particular work, it shows once again that radioluminescence imaging is a very efficient method to find alpha contamination in difficult to access areas.

### 3.2. Influence of Am Concentration and Acidity

A dependency of the radioluminescence intensity on alpha active Am-241 and nitric acid concentrations was studied, since it is not clear what exactly causes the light emission. The role of nitric acid was tested with samples, where the activity was held constant by having 150ppm of Am-241 while the nitric acid concentration was varied. The effect of Am-241 was studied with samples, where the ${\mathrm{HNO}}_{3}$ concentration was held constant at 3M while the americium concentration was varied. In all the experiments, the post-processing of the radioluminescence images was done as outlined in Section 2. The radioluminescence signal level was calculated as an average of pixels corresponding the liquid volume of the cuvette. Then this average was corrected for the background level of the image. The resulting data is presented in Figure 6. The radioluminescence shows little or no dependency on the nitric acid concentration. However, the data shows that the radioluminescence is linearly dependent on the Am-241 concentration. Thus, we can exclude nitric acid from potential luminescent species in the solutions. We can further reduce the number of possible sources of radioluminescence by comparing our results with experiments by using an ion beam as an ionizing source [22]. In that work, a water phantom was irradiated with a carbon-ion beam producing radioluminescence with properties similar to what we found. This leads us to conclude that Am-241 is unlikely linked to the luminescent properties in other ways than as a source of the ionizing radiation.

### 3.3. Radioluminescence Spectrum

Radioluminescence properties of the both solid and liquid samples were further studied by measuring the spectra of the luminescence using a monochromator and a photomultiplier tube. The measurements of the solid and liquid samples were otherwise performed in very similar way, except the liquid sample was with an extra quartz glass wall of the cuvette in the optical path and different spectral scanning parameters were used. For the solid sample, only a single scan from 280nm to 500nm in 0.5nm steps was used and the integration time at each step was set to 30s. To record spectrally narrow nitrogen radioluminescence lines in air over the solid sample, we used the grating model 53020 with a groove density of $1800\;\mathrm{gr}\;{\mathrm{mm}}^{-1}$ blazed at 250nm yielding to 1nm spectral resolution. These modifications allowed us to record the well-known radioluminescence spectrum of air, shown in Figure 7a, in less than four hours. For the liquid sample, the procedure followed the steps described in Section 2. The spectral resolution for a liquid sample was 6nm and the scanning was performed from 280nm to 550nm in 3nm steps, integrating for 10s at each step. The spectral scan was repeated overnight for 88 times to achieve reasonable signal-to-noise ratio for the weak liquid radioluminescence resulting total recording time of 22 h. Figure 7b shows the resulted spectrum of the aqueous solution under ionizing radiation. The correct spectral shape of the well-known radioluminescence of air in Figure 7a verifies that the calibration of spectral response has been successful and that the spectral shape of the liquid radioluminescence in Figure 7b can be trusted. The only additional elements in the optical path are the quartz cuvette and the water column, which both have negligible absorption at the measured wavelength range from 280nm to 550nm.

The normalized spectrum of the light produced by the solid sample in Figure 7a shows the $N_{2}$ lines [16], which is commonly associated with radioluminescence in air. This is congruent with other works investigating air radioluminescence [1,7,29]. To demonstrate the validity of the measurement, the spectrum has been superimposed with the normalized spectrum of radioluminescence in air which has been measured at a higher spectral resolution [19]. Contrary to the radioluminescence of air, the spectrum of aqueous solution under ionizing radiation has not been fully resolved earlier apart from a rough estimation using several long-pass filters [22]. Thus, the spectrum of the radioluminescence of aqueous solution in Figure 7b is, to the best of our knowledge, reported here for the first time. The radioluminescence is weak and spectrally broad, which makes it difficult to record. The spectrum shows a broad maximum between 350nm and 400nm with a gradual decrease until 550nm at the longer wavelength side. On the other hand, the spectrum exhibits a pronounced decrease at shorter wavelength side reaching down to the background level already at 300nm. This steep decreasing slope at shorter wavelength side contrasts with the earlier study done with long-pass filters [22], where they estimated a continuous increase towards the shorter wavelengths resembling the spectral shape of the Cerenkov light. Cerenkov light has a spectrum that is proportional to $\lambda^{-2}$, where λ is the wavelength of the emitted light [30]. The data in Figure 7 (b) clearly shows this is not the case with the radioluminescence of aqueous solutions, as there is the steep decrease at short wavelength side of the spectrum. Furthermore, alpha particles are not capable of producing Cerenkov light directly or through secondary mechanisms [31] and therefore we conclude that the measured radioluminescence must have an origin other than Cerenkov radiation. The spectral response of the measurement system is compensated in the results and the air radioluminescence spectrum with the same setup demonstrates the correct spectral shape down to 300nm, thus validating the spectral response of the measurements. The only optical difference between the measurement arrangements is the quartz cuvette, which has a flat spectral transmittance over the whole measurement range and therefore no effect on the results. The steep decrease at the short wavelength side of the spectrum makes it uncharacteristic for broadband emissions such as Cerenkov light and bremsstrahlung emission. We rather speculate that some of the radiolysis products [32] of water, such as $H_{2}O_{2}$, OH or $H_{2}$, might be responsible for the recorded luminescence of water under the ionizing radiation.

## 4. Conclusions

We studied the radioluminescence produced by alpha active Am-241 dissolved into an aqueous solution of nitric acid. We showed that the amount of light created is about 150 times smaller than what is produced with same alpha activity by the radioluminescence of air. The low amounts of light made it necessary to use photon counting devices and intensified cameras to record the data. We analyzed the spatial distribution of the radioluminescence showing a very evenly distributed light emission. Furthermore, we showed for the first time the aqueous radioluminescence spectrum in the wavelength range between 280nm and 550nm, where we found a broad spectrum between 330nm and 500nm. The demonstrated radioluminescence is potentially useful in applications where visible background light can be avoided by light shielding, such as in radioactive liquid monitoring in closed chambers. We concluded that Cerenkov light, bremsstrahlung or other mechanisms producing broadband emission cannot be the source of the observed radioluminescence due to steep decrease at the short wavelength side of the recorded spectrum, which leaves intrinsic processes of water and its radiolysis products as strong candidates for the luminescence.

## Figures and Tables

**Figure 1 sensors-19-01602-f001:**
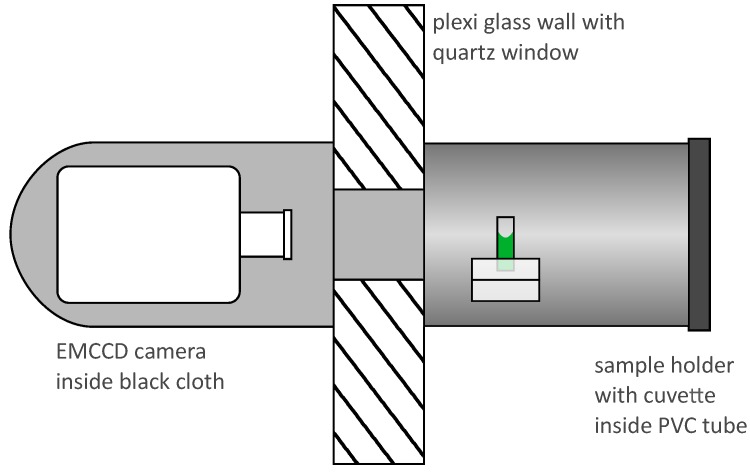
Schematic of the arrangement designed to image radioluminescence: A liquid or solid alpha active sample (green) resides on a holder placed inside black and light-tight PVC tube. The tube is placed around a glove port of box which is holding a quartz glass window instead of a glove. Light exiting the glove box through the window form an image on the sensor of an EMCCD camera. Multiple layers of black fabric wrapped around the camera and the glove port prevent stray light form entering the setup.

**Figure 2 sensors-19-01602-f002:**
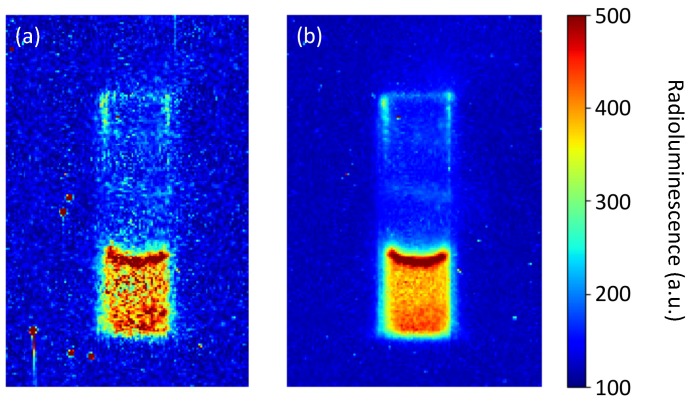
Demonstration of the post-processing using 30 images with exposure times of 60s each. (**a**) A single capture, raw data. The cuvette and the liquid are clearly visible, hot pixels appear as red dots. (**b**) The pixel-wise median over a total of 30 images. The z-scale shows raw pixel values.

**Figure 3 sensors-19-01602-f003:**
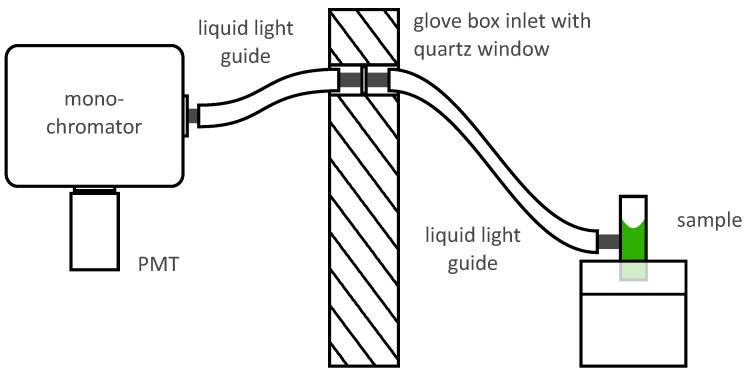
Schematic of the setup designed to measure the radioluminescence spectrum. A liquid or solid alpha active sample (green) resides on a holder inside holder in the glove box. A system of liquid light guides transports radioluminescent light to a monochromator, where it is spectrally separated and detected by a PMT.

**Figure 4 sensors-19-01602-f004:**
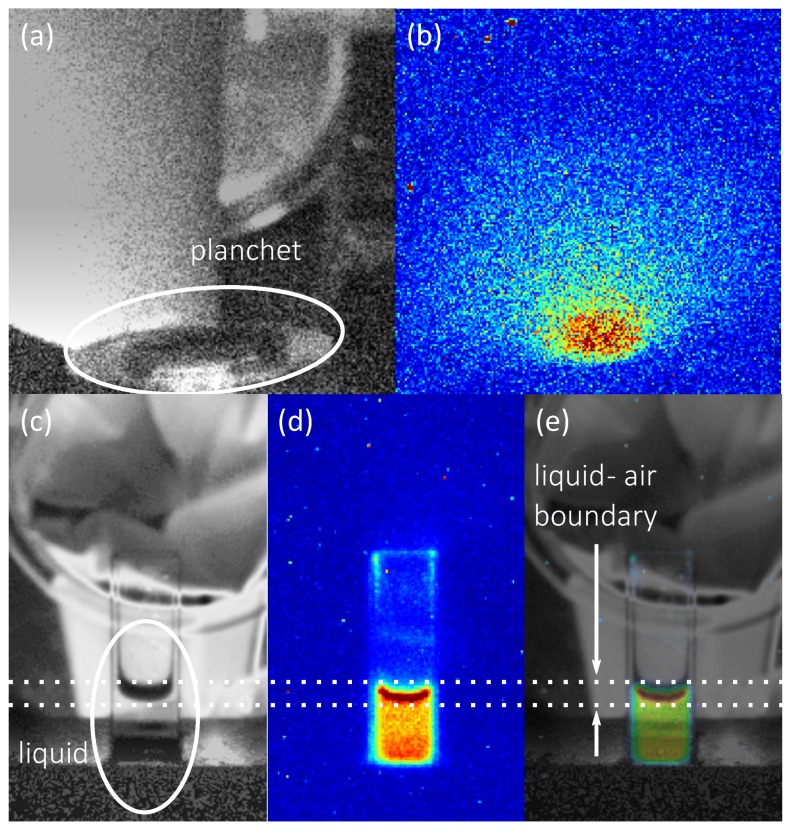
Images of radioluminescence and the locations where they occur. The gray scale images each show a single 0.1s capture of the samples under normal lighting conditions. The colorful images next to them show the same scenes in darkness using much longer exposure times. Dotted lines throughout the images (**c**–**e**) frame the area of the images where the surface of the liquid with air is displayed. The z-scale used in the images is the same as the one used in Figure 2. (**a**) Image of the Pu-239 coated planchet under normal lighting conditions. (**b**) Radioluminescence image of the Pu-239 coated planchet. (**c**) Image of the sample holder and the cuvette which resides in it taken under normal lighting conditions. The camera is slightly tilted and reveals part of the sample holder behind the cuvette. The cuvette holds the transparent liquid, the camera tilt makes it possible to see the boundary of the liquid with the air. (**d**) Radioluminescence image of the cuvette containing 750ppm of Am-241 in 3M
${\mathrm{HNO}}_{3}$ solution. (**e**) Overlay of the cuvette radioluminescence image (**d**) on the image of the cuvette (**c**).

**Figure 5 sensors-19-01602-f005:**
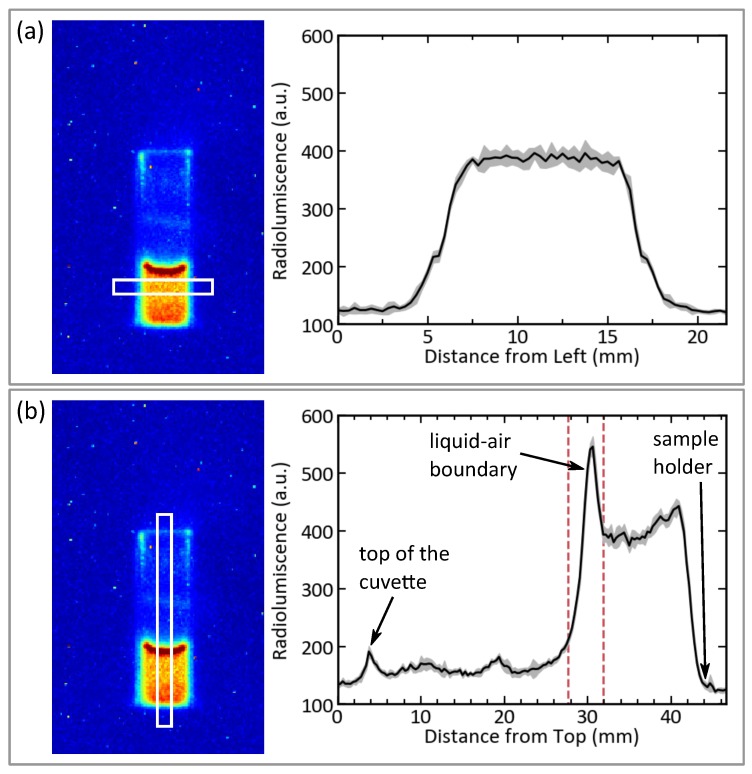
Cross sections of parts of the radioluminescence image of 750ppm Am-241 in 3M
${\mathrm{HNO}}_{3}$. (**a**) Horizontal cross section across the cuvette. The uncertainty displayed in gray is identical to the standard deviation of the pixel values in vertical direction. (**b**) Vertical cross section across the cuvette. The uncertainty displayed is identical to the standard deviation of the pixel values in the horizontal direction. Labels point to notable features and name the position where they occur.

**Figure 6 sensors-19-01602-f006:**
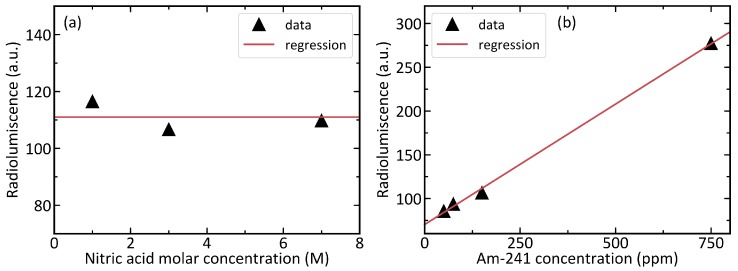
Dependence of the measured radioluminescence intensity on the nitric acid concentration and the americium concentration. The pixel values displayed as radioluminescence have been corrected for the image background. (**a**) Change of the radioluminescence with a varying nitric acid concentration. A constant has been fitted to the data. (**b**) Change of the radioluminescence with varying Am-241 concentration. A linear function has been fitted to the data.

**Figure 7 sensors-19-01602-f007:**
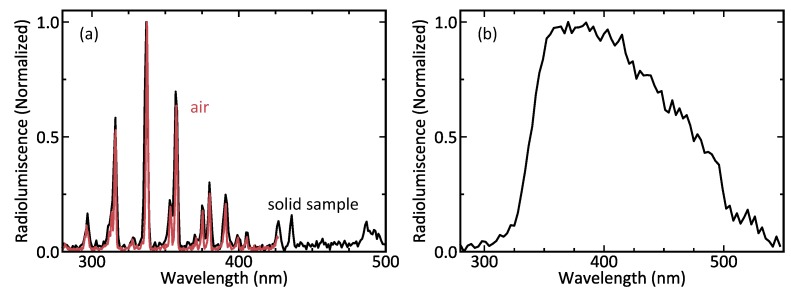
The normalized radioluminescence spectra of (**a**) light created by the solid sample (black) contrasted with this of air (red, data from [19]) (**b**) 750ppm Am-241 in 3M
${\mathrm{HNO}}_{3}$ solution.

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
