# Peer review of "Alpha Radiation-Induced Luminescence by Am-241 in Aqueous Nitric Acid Solution"

_sensors, 2019, doi:10.3390/s19071602_

Reviewer 1 Report

Very nice article, which requires just a little polishing of the presented results.

Why was HNO3 selected? Easiest liquid with nitrogen?

Line 94 definition of molar concentration

Line 110 distance camera/sample?

Line 123 why blurred first. Why just don’t sum first, hot pixels will be removed by summing if not the same for all pictures? The edge effect seems to be result of the blurring. Don't see motivation for the process as described with blurring first.

Figure 2 is missing scale on Z axis

Figure 4 a) very bad quality why? c), d) different scales on z-axis? Should show it.

Figure 5 scale on y axis, at least to be able do some relative estimates, now difficult to see difference between plume, middle and bottom of the liquid

Figure 6 y axis again no scale. 6a) there seems to be slope, while you claim no dependence on the HNO3 concentration. Fit is also very bad middle point out of line completely. Wouldn’t one think that higher concentration of HNO3 would lead to more N atoms being excited by each alpha? But maybe the reach in the liquid is also concentration dependent. This requires little more discussion, I think.

Author Response

Responses are in the attached pdf.

Reviewer 2 Report

The measurement is interesting, but a lot of laboratory work is still necessary to publish work.

A reasonable data analysis is missing, calibrations are missing and the paper conclusions are not supported by the sole measurements.
Therefore it is not clear to me what the innovative results of this paper/measurement are in this status.

As first I have a comment on the data handling.To remove the hot pixels due to 59.6KeV of Am-241 the image are "median blurred with a sliding window width of 7".
This results in a very poor definition images that affects the subsequent analysis capability.
However there is a very simple way to recognize and remove these hot pixel without any pixel blurring, since you have 30 different images.
I suggest to do the pixelwise average, averaging pixel-by-pixel removing the largest,  that is averaging the lower 29 values of the same pixel and always removing the value with the largest measurement (that could be the X-ray).
This unblurred-cleaned image would improve a lot the quality of the result.

Here I summarize the statement/result that are in the paper and that are not fully convincing me:
1) "The light production in the liquid is evenly distributed throughout the sample volume having a slight increase on the surface of the liquid."
This statement in the result of the discussion of fig. 5.b.
-There a luminosity broad peak is labelled as "liquid bottom",
-another peak is labelled "plume"
-and 3 small peaks are labelled ad "contamination".
The peak labelled as "liquid bottom" is ascribed to be the "result of reflection from the cuvette holder".
This could be reasonable, since the optical effect of the top of the cuvette border is also clearly visible
as two side spots also in the 2D color plot (left) of fig 5. (or also better in unblurred fig 2.a)
and this structure is producing the first peak of fig 5.b (right) (distance 2mm from top) that is wrongly labelled as "contamination".

The "plume" peak is instead ascribed to a physical effect: "The radioluminescence emanating from the cuvette in Fig. 4 (d)
appears to be more intense at the surface of the liquid. This might be due to alpha particles leaving the
liquid into the air and depositing their energy there, thus leading to more efficient radioluminescence production in air"
This would support the other conclusion of the paper that alpha luminescence in air is more efficient that in liquid.
However this interpretation of the plume is not convincing me, since, similarly to the first peak of "contamination" it could be that

the "plume" is an optical effect of the liquid meniscus, that is clearly
visible in fig. 4.b.

To solve this problem and to give solidity to the conclusions, I suggest to compare this measurement with another
control one using a some known luminescent liquid (with known spectra).
Moreover I suggest to not use blurring to improve the geometry information in these figures (as suggested above)
and I suggest also to superimpose to fig 5 a line-drawing of the cuvette border and of the curved liquid meniscus
and to give the same information to the left plots of fig.5 adding more vertical lines
(there only liquid surface is now shown). Moreover I would see superimposed to fig. 5 right plots
2 more plots of other slices (scaled to fit same height): e.g. fig. 5a a slice outside the liquid and a slice
with liquid meniscus fig 5.b also left and right slices/band near the central one
that is shown now.

2) Radioluminescence spectrum.
The shape of fig 7.b is analyzed to infer that:
"The steep decrease at the short wavelength side of the spectrum makes it unlikely that the observed light is Cerenkov light, bremsstrahlung or any other similar broadband emission." It is not clear to me how the absorption coefficient of
water/cuvette and readout system have been accounted for. In particular in liquid water a very strong absorption is expected
exactly below 400nm as measured. Also in this case I would expect to see the correction for these absorption effects or to see the comparison/calibration of the measured spectra with a reference one using the same experimental set-up.

Without similar quantitative analysis is meaningless the inference about the physical mechanism responsible for the observed luminescence.

Author Response

Responses are in the attached pdf.

Reviewer 3 Report

In general, this work presents interesting findings and with some alterations would be suitable for publishing.

The research is presented in a clear way in the main and results are easy to see. The work does offer a novel element, which is interesting and useful to an understanding of the phenomenon involved.

However, modifications are required to show a complete picture in terms of the reference to other research. Work in the UVC wavelength range (180-280 nm) appears to have been ignored by the authors, and conclusions based on results from only the 300 – 400 nm wavelength range are presented. For example, line 1 of the abstract states that radioluminescence is caused by interaction with nitrogen in air, where in the UVC wavelength range it is generally accepted that this is not the case. As the work carried out in this paper in general does not include UVC wavelength range, it would seem sensible not to go into detail. However, conclusions from work done only in the  UVA and UBV wavelength range, which may not apply to the UVC, do not show the whole picture and are not wholly accurate, for example the effect of background lighting and the radioluminescence mechanism. These should be corrected.

Here are comments using line numbers for reference:

Line 44: It is a bold assertion that Sand et al have ‘answered the question’ of how ‘good’ a detector can be, given the set of circumstances that you list. This would seem to suggest that there is no room for further study. Some changes to the wording is needed to make it clear that this is an answer to the question, but not necessarily the only possible answer.

Line 50: In this section it states that background lighting means that detecting radioluminescence isn’t feasible. Reference to the work of Crompton et al. (2017, 2018) and Ivenov et al. (2009, 2011) would show that in the UVC wavelength range there has been positive results in detecting radioluminescence without interference from background light. Both these authors are conspicuous by their absence from your references. Although it is stated that the effect of background lighting on radioluminescence is not the purpose of the paper, it is important to point out that it is only not feasible in the 300 – 400 nm wavelength range due to background light. Although there is a lot of positive work in the UVC wavelength range it has been dismissed with this comment as it stands.

Line 56: The recent work of Crompton et al. (2018) on radioluminescence in gasses in the UVC wavelength range has been missed out here.

Line 82: An indication of the transmittance of the quartz window by wavelength would be useful here.

Line 98: Says ‘…samples divided on 2 distinct sets…’ should this be ‘… into 2 distinct sets’?

Line 239: Some indication of the effect of wavelength here would be useful if available. As it is so important in the rejection of background lighting, it is important to clarify wavelength ranges where possible.

Author Response

Responses are in the attached pdf.

Round  2

Reviewer 2 Report

The draft and data quality has much improved now

however the conclusions are still not convincing
and they are not quantitatively supported.

Comments in attached file.

Author Response

Responses in the attachement
